# Pompe Disease: New Developments in an Old Lysosomal Storage Disorder

**DOI:** 10.3390/biom10091339

**Published:** 2020-09-18

**Authors:** Naresh K. Meena, Nina Raben

**Affiliations:** Cell and Developmental Biology Center, National Heart, Lung, and Blood Institute, NIH, Bethesda, MD 20892, USA; nareshkumar.meena@nih.gov

**Keywords:** Pompe disease, lysosome, lysosomal targeting, autophagy, enzyme replacement therapy, gene therapy, muscle, satellite cells

## Abstract

Pompe disease, also known as glycogen storage disease type II, is caused by the lack or deficiency of a single enzyme, lysosomal acid alpha-glucosidase, leading to severe cardiac and skeletal muscle myopathy due to progressive accumulation of glycogen. The discovery that acid alpha-glucosidase resides in the lysosome gave rise to the concept of lysosomal storage diseases, and Pompe disease became the first among many monogenic diseases caused by loss of lysosomal enzyme activities. The only disease-specific treatment available for Pompe disease patients is enzyme replacement therapy (ERT) which aims to halt the natural course of the illness. Both the success and limitations of ERT provided novel insights in the pathophysiology of the disease and motivated the scientific community to develop the next generation of therapies that have already progressed to the clinic.

## 1. Introduction

Pompe disease, also known as Type II glycogen storage disease (GSDII), is a rare autosomal recessive neuromuscular disorder that affects people of all ages. This severe, often fatal illness gained a well-deserved reputation for being the first recognized lysosomal storage disorder, a group which now includes more than 50 entities [1]. In the 1960s, some thirty years after the first description of the illness by Joannes Cassianus Pompe [2], the mystery behind massive accumulation of glycogen in multiple tissues in autopsy reports from the affected individuals was solved: the patients were missing the enzyme, acid alpha-glucosidase (GAA), which had an acidic pH optimum, and glycogen was stored in a membrane-bound organelle, suggesting its lysosomal origin [3]. This enzyme is uniquely responsible for total hydrolysis of glycogen to glucose in the lysosome, and its deficiency manifests as a multisystem disorder with predominant involvement of skeletal and cardiac muscles.

The estimated frequency of the disease is often cited as 1 in 40,000 live births [4], but recent implementation of newborn screening (NBS) for Pompe disease revealed a much higher frequency [5,6]. The first such a program was enacted in Taiwan in 2005 [7], followed by several other countries including the US, where the recommendation by the Advisory Committee on Heritable Disorders in Newborns and Children to add Pompe disease to the Recommended Uniform Screening Panel was finally approved in 2015.

The enzyme is synthesized as a 110-kD precursor, which is glycosylated in the endoplasmic reticulum (ER) and phosphorylated on mannose residues in Golgi apparatus on its way to the late endosome/lysosomes, where the enzyme undergoes extensive proteolytic processing yielding, through a 95-kD endosomal intermediate, fully processed 76- and 70-kD mature lysosomal forms with higher affinity for the natural substrate, glycogen [8,9]. These posttranslational modifications, common to many lysosomal hydrolases [10], ensure their transport to the lysosomal system through the mannose-6-phosphate (M6P) receptor-mediated pathway [11]. Importantly, a portion of the precursor protein can be secreted, taken up by ubiquitous cell surface M6P receptors, and delivered to the lysosome through the endocytic pathway, thus providing the basis for enzyme replacement therapy for Pompe and other lysosomal storage disorders. The structure of recombinant human GAA has been solved by X-ray crystallography [12].

The severity of the disease largely corresponds to the nature of the genetic defect and the degree of residual enzyme activity, giving rise to a wide range of phenotypes that differ in the age of onset and the rate of progression [13,14]. Although this condition represents a single disease continuum, two discrete phenotypes are broadly accepted: the most severe and rapidly fatal infantile onset form and a milder late onset form with or without cardiac involvement respectively.

More than 500 mutations have been reported, including insertions, deletions, splice site, nonsense, and missense mutations (pompevariantdatabase.nl). Most are unique to families, but some are common in particular ethnic groups, such as the Dutch, Chinese from Taiwan, African-Americans, etc., (reviewed in [15]). The most common defect in Caucasians with late-onset disease is the leaky intronic mutation (c.-32-13T>G IVS1) [16] which allows for the generation of low levels of normal enzyme [17,18,19]. A recent study demonstrated that in a subset of patients, the IVS1 mutant allele contains a genetic modifier (c510C>T; a synonymous mutation) which further reduces the amount of active enzyme [20]. The advances in new molecular techniques, such as a generic-splicing assay, minigene analysis, SNP array analysis, and targeted Sanger sequencing, significantly expanded the scope of genetic diagnostic analysis in Pompe disease patients [21].

Following years of drug development, preclinical studies in a mouse model [22,23], and two pivotal clinical trials [24,25], recombinant human GAA-a drug called alglucosidase alfa (rhGAA; Myozyme© (ex-US) and Lumizyme© (US); Genzyme, Cambridge, Massachusetts)-received broad-label marketing approval for the treatment of Pompe disease in 2006. Enzyme replacement therapy (ERT) became the first disease-specific treatment and remains the only approved therapy for Pompe disease. In this review we will explore how the therapy changed the disease landscape by altering clinical picture, stimulating efforts to better understand the pathogenesis, and motivating industry and academia to develop next-generation treatments.

## 2. An Expanded Set of Clinical Characteristics

Pompe disease presents with a wide range of clinical characteristics and different life expectancy depending on the age at onset. Two major distinctive clinical phenotypes are recognized based on the age at which the symptoms appear and the presence or absence of cardiomyopathy: the most severe classic infantile onset type (IOPD) includes patients with less than 1% of GAA activity who develop symptoms within the first year of life and, if left untreated, rarely survive beyond 18 months; the milder late-onset type (LOPD) with higher enzyme activity may become apparent in childhood, adolescence, or adulthood.

Patients with IOPD present with rapidly progressive hypertrophic cardiomyopathy, left ventricular outflow obstruction, elevated creatine kinase, generalized muscle weakness, hypotonia, macroglossia, failure to thrive, respiratory distress and loss of independent ventilation. Major motor milestones, such as the ability to roll over, sit, or stand, are not achieved; the majority of patients die from a combination of heart and respiratory failure. Although this most devastating type is clinically homogeneous, there is an important distinction among the patients in that some produce non-functional enzyme (cross-reactive immunological material (CRIM)-positive), whereas others produce no enzyme at all (CRIM-negative). Additionally, a subset of patients with similar age of onset and clinical presentations but without/or less severe cardiomyopathy and absence of left ventricular outflow obstruction are classified as non-classic IOPD; delayed motor skills and severe progressive muscle weakness leading to respiratory failure by early childhood are main characteristics of the disease in this subgroup [26,27].

The introduction of ERT has been a major breakthrough in the field, and the treatment proved to be most beneficial for patients with IOPD who survive significantly longer due to a remarkable effect of the drug on cardiac size and function. The reversal of hypertrophic cardiomyopathy and the improvement of cardiac function were observed in most patients within the first months on ERT. This effect of the drug was already evident in the first clinical trials [24,25], and the expectation was that the treatment would convert the most severe infantile form to a milder late-onset type. However, in more than a decade since the regulatory approval of the drug, it became clear that this is not the case. The survivors, in particular long-term survivors develop a new phenotype that reflects multisystem involvement with some of the symptoms that have not been previously ascribed to Pompe disease [28]. Almost all long-term survivors are plagued with muscle weakness despite ERT, leading to a characteristic gait pattern, lumbar hyperlordosis, and progressive scoliosis. Respiratory dysfunction that results from both muscular and neural deficits [29] is a common problem necessitating non-invasive or invasive assisted ventilation. Hearing loss, ptosis, a motor speech disorder, and much reduced speech intelligibility are all manifestations of a new emerging phenotype [28] (reviewed in [30]). Additional symptoms include swallowing difficulties, gastro-esophageal reflux, and aspiration leading to increased risk of upper respiratory tract infections and pneumonia. Accumulation of excess glycogen in the brain of patient with IOPD [31] may lead to neurocognitive impairment in a number of patients [32]. A recent study [33] reported progressive white matter lesions which affect cognitive and neuropsychological development in ERT-treated patients with IOPD who reached adulthood. These changes were not seen previously because of early fatality in untreated IOPD patients.

Several basic conclusions can be drawn: A number of patients still die within the first years of life despite ERT; the response to ERT in CRIM-negative patients is confounded by an immune response against the enzyme [34] and is less effective even with immunomodulation; the best results are achieved when the patients are afforded very early treatment-within the first few days of life-which became feasible with the initiation of newborn screening [35]. However, even under the best-case scenario, when ERT was initiated within the first 10 days of life in CRIM-positive IOPD patients, airway abnormalities and speech disorders were still observed over the course of 8 years in a retrospective study [36]. Finally, a new phenotype with clinical manifestations of the CNS lesions requires additional intervention since the neurological manifestations are not amenable to ERT and raises new questions about how treatment will be shaped in the future.

The symptoms of limb-girdle muscle weakness, respiratory dysfunction, and hyperCKemia at any time beyond 12 months of age are the main characteristics of the late-onset form Pompe disease. Initial signs, such as exercise intolerance, muscle pain, and fatigue, are often overlooked or ignored leading to a delayed diagnosis [37,38]. Later on, slowly progressive deterioration of axial, limb-girdle, and respiratory muscle, particularly the diaphragm, eventually leads to wheelchair-dependency and assisted ventilation. The lower airway smooth muscles involvement (well documented in a mouse model [39]) contributes to respiratory problems, and respiratory failure is the main cause of death in LOPD even in patients on ERT.

The view of LOPD has shifted away from its original definition as primarily a muscle disorder. As in IOPD, the expanded spectrum of clinical manifestations reflects the involvement of multiple tissues and organs, thus establishing the multisystem nature of this illness. The emerging new features of the disease include vascular and cerebrovascular defects, for example, intracranial aneurysm or dilative arteriopathy, hearing loss, central and peripheral nervous system abnormalities, scoliosis, progressive loss of bone mass, as well as gastrointestinal and urinary tract symptoms [40] (reviewed in [41]). A small subset of LOPD patients develop cardiomyopathy which improves on ERT [42].

The majority of LOPD patients benefit from ERT in that the therapy improves or stabilizes motor and respiratory function but skeletal muscle weakness persists, and many show signs of disease progression [43,44]. Inefficient glycogen clearance in smooth muscle of vascular, ocular, gastrointestinal, and respiratory systems has been reported in ERT-treated patients (reviewed in [45]). A number of patients still require ventilation and become wheelchair dependent despite several years on ERT [46]. A recent long-term (10 years) prospective study reported that the initial positive response to therapy in the first 3–5 years was followed by a secondary decline in walking ability, muscle strength, and pulmonary function [47].

Overall, there is a great deal of variability in LOPD patients’ response to therapy, ranging from those who continue to respond well up to 8–10 years to those who do not respond at all. Although the reasons of this variability are not exactly understood, the extent of underlying muscle pathology is, no doubt, an important factor. Not surprisingly, a detailed analysis of muscle biopsies from a large cohort of LOPD patients before the ERT initiation demonstrated a correlation between the severity of muscle damage and response to therapy. However, no correlation between the duration of the disease and more severe muscle damage was observed, and there was no meaningful genotype–phenotype correlation in this group [48].

The lack of any clinical stabilization or improvement in the first 2 years of treatment or deterioration of clinical condition despite therapy constitute the reasons to consider the discontinuation of ERT; these criteria are included in the guidelines developed by a multidisciplinary expert committee of the European Pompe consortium (EPOC). Interestingly, no acceleration in decline was observed in several patients who discontinued ERT in accordance with the established criteria [49,50], suggesting that at least in some cases ERT is of dubious benefit. As for the timing to start ERT, the European consensus states that only symptomatic patients with confirmed diagnosis should be treated. However, this practice may not last long because LOPD patients, diagnosed through new-born screening and carefully followed up, were shown to have overt or subtle symptoms much earlier than expected [51]. The search for reliable biomarkers to aid in diagnosis, progression, and response to ERT has remained elusive over the past decades. In this regard, of particular interest is the recent identification of circulating microRNAs as potential biomarkers of Pompe disease [52,53] (reviewed in [54]). These new biomarkers, in particular miR-133a, can help monitor the efficacy of the current and emerging therapies by serial sampling through liquid biopsies.

## 3. Beyond the Lysosome: Pathogenic Cascade and Muscle Regeneration

Our understanding of the mechanisms driving the pathogenesis of muscle damage in Pompe disease is rapidly evolving. Not long ago, the pathogenesis was viewed as a process that occurs in stages, including gradual glycogen buildup in the lysosomal lumen, lysosomal membrane rupture due to the mechanical pressure, a discharge of glycogen and potentially toxic materials into the cytoplasm, and finally the destruction of muscle architecture [55]. These somewhat loosely defined morphological stages as well as normal looking myofibers next to the affected ones, can indeed be recognized in a biopsy from the same LOPD patient. This heterogeneity in the degree of myofibers damage within a single muscle bundle is one of the mysteries in Pompe disease. In contrast, muscle morphology in IOPD patients is more homogeneous, normal fibers are commonly absent, and muscle architecture is lost in most cells.

The first “blow” to this simplistic view of the pathogenesis came from the experiments in GAA KO mice (KO) [56]. Ultrastructural and morphological observations revealed the presence of large areas of autophagic accumulation in muscle samples from KO [22]. Following this initial finding, we began to unravel the contribution of autophagy to the pathophysiology of Pompe disease.

Macroautophagy (often referred to as autophagy) is a major degradative pathway that delivers cytoplasmic cargo to the lysosome where the final breakdown occurs. Various types of cargo including a portion of the cytoplasm, misfolded protein aggregates, glycogen, and in the case of eukaryotes, malfunctioning organelles, are transported to the lysosome in newly formed double membrane-bound vesicles, called autophagosomes, that fuse with lysosomes. This process requires the recruitment and assembly of multiple components of the autophagy machinery. Basal autophagic activity maintains tissue homeostasis and serves as quality control mechanism by promoting the degradation of toxic protein aggregates and aberrant organelles [57]. Aside from its basal role, autophagic activity can be induced in response to a variety of stress conditions, providing an adaptive mechanism to ensure the cell’s survival [58,59].

Skeletal muscles rely heavily on autophagy because muscle fibers are terminally differentiated cells unable to divide and dilute aberrant proteins and organelles through cell division [60]. It is now well-established that too much or too little autophagic activity can negatively affect muscle function and result in muscle wasting [61,62]. Accumulation of autophagic debris is a prominent feature of a group of muscle disorders, including Pompe disease, called autophagic vacuolar myopathies [63].

The sheer size of the areas occupied by the piles of autophagic debris in muscle fibers in KO mice is quite remarkable. Long stretches of autophagic accumulation, often extending the whole length of the fibers and located in their cores, can be visualized by confocal microscopy of single muscle fibers immunostained with lysosomal (LAMP1) and autophagosomal (LC3) markers. Ultrastructural studies revealed the contents of the autophagic areas: classical double-membrane autophagosomes with undigested materials or glycogen particles, glycogen-laden lysosomes with broken borders, multivesicular bodies, and multimembrane concentric membranous structures. These areas are completely devoid of sarcomere structure and, therefore, are unable to contract. Indeed, there is a significant reduction in muscle force ex vivo in KO compared to WT [64].

The molecular mechanism underlying the defective autophagy in the diseased muscle involves both increased formation of autophagosomes (induction of autophagy) and their inefficient fusion with lysosomes (autophagic block). Elevated levels of proteins required for the initial steps of autophagy (autophagosome nucleation and maturation), such as VPS15 protein kinase, the lipid kinase catalytic subunit VPS34, and the regulatory protein Beclin1, in muscle biopsies from KO and Pompe patients [65,66] argue for autophagy induction. The mechanism of this surge in autophagy is not exactly clear. Likewise, the mechanism of impaired autophagosomal-lysosomal fusion, which can be directly observed by time-lapse microscopy of live fibers co-stained with LC3/LAMP1 [67], is not fully understood. Our recent finding of increased levels of galectin 3, a sensitive marker of endosomal/lysosomal damage, in KO muscle offers a possible explanation for the defective fusion [66].

Defective autophagy, a major secondary abnormality in the affected muscles, can have dire consequences: oxidative stress, accumulation of aberrant mitochondria and autophagic substrates, such as p62/SQSTM1 and potentially toxic high molecular weight K63-linked ubiquitinated protein aggregates. Furthermore, autophagic buildup negatively affects trafficking and lysosomal delivery of the therapeutic enzyme [67,68,69] and contributes to poor skeletal muscle response to ERT [70,71]. Importantly, autophagic accumulation and elevated levels of galectin 3 can be detected as early as in 6-week-old KO mice, well before the disease becomes clinically apparent [66].

Yet another layer in the pathogenic cascade of muscle damage in Pompe disease involves dysregulation of lysosome-based signaling pathways. Gone is the view of the lysosome as a low-key cellular recycling center. New research over the past decade positions this organelle at the center of metabolic signaling pathways. The lysosome integrates multiple environmental signals to maintain cellular homeostasis by regulating the switch between anabolic and catabolic state. Two major kinases that have broadly opposing effects-the nutrient-sensitive mammalian target of rapamycin complex 1 (mTORC1) and the energy-sensing AMP-activated protein kinase alpha 1 (AMPK)-play principal roles in controlling metabolic programs. Much is now understood about the role of the lysosome as a platform for the activation of these kinases, and how these two signaling pathways cross-talk directly and indirectly at multiple levels to balance ATP-consuming biosynthetic and ATP-producing catabolic pathways and to send feedback signals for lysosomal biogenesis and autophagy [72]. The communication and coordination between the mTORC1 and AMPK pathways involve an ever-growing number of signaling intermediates, but this field is beyond the scope of this review. Here, we will name only those few that have been analyzed in Pompe disease.

The recruitment of mTORC1 to the lysosomal membrane under nutrient-rich condition brings the kinase to close proximity to its activator, Ras homologue enriched in brain (RHEB), which resides on lysosomes. RHEB is negatively regulated by the tuberous sclerosis complex (TSC2) leading to the inhibition of mTORC1 activity. In its turn, TSC integrates a range of inputs including AMPK pathway [73,74] (reviewed in [75,76,77,78]). Activation of AMPK by upstream liver kinase B1 (LKB1) [79] occurs on the surface of late endosome/lysosome, where AMPK phosphorylates and activate the TSC complex, thereby inactivating RHEB and mTORC1 signaling [76]. When nutrients are plenty, mTORC1 phosphorylates and inactivates a key initiator of autophagy, ULK1, on Ser758 to inhibit autophagy and prevent its interaction with AMPK [80]. On the other hand, AMPK phosphorylates ULK1 on multiple different sites (e.g., Ser317 and Ser777) to activate ULK1 and induce autophagy [80,81].

The dysregulation of AMPK and mTORC1 signaling pathways and defective autophagy have been linked to changes in muscle function and disease [61,82,83] (reviewed in [84]). Extensive analysis of the upstream inputs and downstream targets of AMPK and mTORC1 in cultured GAA-deficient myotubes and in muscle samples from KO mice, performed in our lab, demonstrated a dysregulation of these two pathways in the diseased muscle cells. Increased levels of LKB1 and an increase in the amount of LKB1-mediated Thr^172^ AMPK phosphorylation indicated activation of AMPK, suggesting that the affected muscles are energy deficient. Increased phosphorylation of two downstream AMPK targets-acetyl CoA carboxylase (ACC Ser^79^) and TBC1D1 (Ser^660^)-confirmed the activation of the LKB1/AMPK pathway. In line with this notion, AMPK-mediated phosphorylation of TSC2^S1387^ and ULK-1^S317^ were markedly increased in KO muscle, whereas mTORC1 activity was decreased as measured by the phosphorylation levels of its downstream targets, EIF4EBP1 and the p70 ribosomal protein S6 kinase [85,86]. What is intriguing about the mTORC1 status in the affected muscle is that this kinase is able to properly move to the lysosome under nutrient-rich condition but unable to move away from the lysosomal surface under starvation, as shown by immunostaining of GAA-deficient myotubes with LAMP1/mTORC1 [85]. The disturbance of mTORC1 signaling was also reported in GAA-deficient C2C12 myoblasts, and in human fibroblasts and induced pluripotent stem cells (iPSCs) from infantile Pompe disease patients [87,88]. Thus, it appears that activation of the LKB1-AMPK pathway and excessive accumulation of TSC2 at the lysosomal surface are responsible for the diminished mTORC1 activity and activation of autophagy in Pompe muscle.

In developing and growing muscle, mTORC1 is viewed as a key regulator of skeletal muscle mass [89]. Indeed, reactivation of mTORC1 in vivo in muscle of KO mice by AAV-mediated TSC knockdown or arginine supplementation resulted in the reversal of muscle atrophy and a striking removal of autophagic buildup [85]. A similar effect was observed in a model of another muscle disorder: activation of mTOR ameliorated muscle atrophy in valosin-containing protein associated inclusion body myopathy (VCP-IBM) [90].

Finally, our recent studies demonstrated a metabolic re-programming in muscle samples derived from KO mice. The metabolome profile of the diseased muscle reflected the state of limited glucose availability and revealed a decrease in glycolysis and a shift from carbohydrate to lipids as the main energy source. Lower than normal glycolysis was also observed in human primary myoblasts from Pompe disease patients [91]. It is worth mentioning that metabolic pathways are highly conserved through evolution, and metabolic similarities between rodents and humans are very comparable despite the commonly observed differences in the phenotype of many mouse models of human diseases, including Pompe disease. In addition, a consistent increase in glycogen synthesis precursors in KO muscle-galactose 1-phosphate and UDP-glucose, the immediate glucose donor for glycogen synthesis-suggests inhibition of cytosolic glycogen synthesis [92]. Thus, progressive lysosomal glycogen accumulation in the diseased muscle sets off a cascade of events, such as altered autophagy and muscle proteostasis, oxidative stress, and dysregulation of major signaling and metabolic pathways (Figure 1). Despite a significant progress in our understanding of the molecular mechanisms of muscle damage in Pompe disease, we are still far from having the whole picture [93].

A somewhat disappointing result of ERT in reversing skeletal muscle pathology became a driving force behind yet another new direction in the field, namely exploring the regenerative capacity of skeletal muscle in Pompe disease. Muscle wasting and atrophy are not only the result of increased breakdown of damaged muscle cells, but also a decreased ability of satellite cells to replace lost myofibers (reviewed in [94]). Activation of Pax7-positive multipotent satellite cells is a prerequisite for muscle regenerative response [95] (reviewed in [96]). These cells, located between the sarcolemma and basal lamina of myofibers, typically exist in adult muscle in mitotically quiescent state, but when activated begin to proliferate and differentiate, leading to the replenishment of the quiescent satellite cell reserve and the formation of new myofibers for muscle repair.

Importantly, analysis of muscle biopsies from both IOPD and LOPD patients indicated that the pool of Pax7-positive satellite cells was well preserved in each group independent of the disease severity, but the regenerative activity of muscle was absent; mild regenerative response was detected only in some classic infantile patients [97]. These data provided indirect evidence of a failure of satellite cell activation in Pompe disease. In the follow-up studies in murine models of Pompe disease, two independent groups [98,99] demonstrated that despite extensive muscle damage, the number of regenerating fibers in adult animals is essentially negligible. Remarkably, the satellite cells were fully functional and were able to robustly respond to acute and repeated muscle injury induced by cardiotoxin or barium chloride leading to efficient muscle regeneration comparable to that in WT controls. Thus, muscle from GAA-KO mice regenerated efficiently in response to exogenous insult but failed to do so to counteract relentless damage during the course of the disease. Resolving the apparent conundrum of why muscle SCs remain inactive in Pompe disease, while still retaining their potential to become activated, may open up new therapeutic venues. Given the role of autophagy in providing nutrients needed to meet the bioenergetic demands during transition of satellite cells from quiescence into an activated state [100], defective autophagy in the diseased muscle may be responsible for the missing activation signal.

## 4. Evolution of Therapy

There is no doubt that Pompe disease patients have benefited from alglucosidase alfa treatment. Although it is clear that the current drug is no panacea, the therapy has added years and quality of life to the lives of people with the disease. It is also equally clear that a more efficient therapy is much needed. The number of studies designed to improve the therapy, both in vitro and in vivo in mouse models, has skyrocketed in past years. Multiple approaches, such as substrate reduction therapy, inhibition of autophagy and modulation of mTORC1 signaling, chaperone therapy, stimulation of lysosomal exocytosis, antisense oligonucleotides, etc., have been explored as a potential alternative or adjunct therapies. We refer the reader to several recent reviews on these subjects [101,102,103]. Instead, we will cover the strategies that have already moved to the clinic and focus on the completed, ongoing, or enrolling clinical trials evaluating the effects of new recombinant enzymes and gene therapy.

A major shortcoming of the current standard of care is the inability of alglucosidase alfa to reach skeletal muscle efficiently; in fact, less than 1% of i.v. bolus administered enzyme ends up in muscle [104]. As mentioned above, the cation-independent mannose 6-phosphate receptor (CI-M6PR) has been exploited to deliver the exogenous recombinant enzymes for the treatment of lysosomal storage disorders. The receptor binds its ligands, the M6P bearing proteins, at the cell surface and transports the cargo to acidic late endosomal compartment, where the proteins and the receptor part ways, so that the proteins move to the lysosome whereas the receptor is recycled for the next round of ligand transport. A major reason for the suboptimal efficacy of alglucosidase alfa in skeletal muscle is that M6P-enzyme represents only a small fraction of the drug since its overall content of M6P glycan is low [23]. The problem is further compounded by the relatively low abundance of CI-MPR in muscle tissue [105,106].

### 4.1. Next-Generation ERT

Notably, the less than optimal intrinsic quality of alglucosidase alfa was understood years ago, as indicated by the efforts in the early 2000s to enhance the delivery of the therapeutic enzyme to the affected muscle by carbohydrate remodeling of the original enzyme to increase the amount of M6P [107]. This early work and an extended follow-up study [108] led to the development of a modified glycoengineered enzyme with a synthetic oligosaccharide harboring mannose 6-phosphate (M6P) residues. This new recombinant human GAA, called neo-GAA, with much improved affinity for the CI-MPR and uptake by muscle cells, showed more efficient glycogen clearance in immunotolerized KO mice compared to the unmodified enzyme, particularly in younger mice. Muscle glycogen reduction was also achieved in older symptomatic KO mice, but the improvement in motor function was only marginal [108].

Neo-GAA (Avalglucosidase alfa; Sanofi Genzyme, Cambridge, MA, USA), a second-generation glycoengineered recombinant GAA with increased bis-M6P levels, was first evaluated for safety and tolerability in a now completed clinical trial (NCT01898364). The results of this Phase 1/2 open-labeled, ascending-dose (5, 10, and 20 mg/kg biweekly over the course of 24 weeks) study in previously ERT-treated (switch group) and -untreated (naïve group) LOPD patients was recently published [109]. Neo-GAA was overall well-tolerated and safe; only two of the 24 enrolled patients discontinued because of the drug-related serious adverse events. Based on the glycogen levels in baseline quadriceps biopsies (~6% of tissue area), patients from both groups were considered to be mildly affected. Although the assessment of efficacy was not a part of the protocol, exploratory efficacy parameters showed a slight improvement in pulmonary function in ERT-naïve and no decline in ERT-switch patients. A Phase 3 study to compare the safety and efficacy of neo-GAA and alglucosidase alfa in previously untreated LOPD was initiated in 2016 and is ongoing (COMET; NCT02782741). In the randomized, double-blind portion of the study, patients received either neo-GAA or alglucosidase alfa (standard of care) for 49 weeks; thereafter, all patients participated in ongoing open-label treatment with neo-GAA. In addition, the company’s clinical development program includes several other exploratory efficacy studies with neo-GAA in LOPD and IOPD patients.

Another new investigational drug, AT-GAA (Amicus Therapeutics, Cranbury, NJ, USA) is the combination of a novel non-modified rhGAA bearing high bis-M6P (Amicus proprietary cell line) with a pharmacological chaperone (AT2221; N-butyldeoxynojirimycin; NB-DNJ, miglustat) for stabilizing the enzyme in the circulation (reviewed in [104]). Both preclinical and clinical studies in Pompe disease patients demonstrated that small molecules chaperones increased the stability and bioavailability of the therapeutic enzyme [103,110,111,112].

In a large preclinical study in KO mice, AT-GAA was shown to significantly outperform alglucosidase alfa in all measured outcomes: GAA uptake and activity, muscle strength, reduction in lysosomal size and glycogen levels, and mitigation of autophagic defect [92]. Furthermore, long-term treatment of KO with AT-GAA completely reversed muscle lysosomal glycogen accumulation, eliminated autophagic buildup in >80% of muscle fibers, and to a large degree restored AMPK/mTORC1 signaling, muscle proteostasis, and metabolic abnormalities [66]. This outcome is in striking contrast with the limited effect of a long-term treatment of KO mice with alglucosidase alfa at a similar dose of 20 mg/kg [22].

AT-GAA was evaluated in Phase 1/2 clinical trial in ERT-switch and naïve non-ambulatory LOPD patients; the drug was well-tolerated with a low number of infusion-associated reactions and showed promising results, as evidenced by improvement in muscle function, increase in upper-body muscle strength, and patient-reported outcomes (reviewed in [104]). Two Phase 3 studies, one comparing AT- GAA with alglucosidase alfa/placebo (PROPEL Study; NCT03729362; active, not recruiting), and the second one (ZIP Study: NCT03911505; recruiting) evaluating the pharmacokinetics, safety, efficacy, and pharmacodynamic of AT-GAA in LOPD patients were initiated in 2018 and 2019 respectively.

An attempt was made to target both lysosomal and extra-lysosomal glycogen accumulation in the affected muscle. VAL-1221 (Valerion Therapeutics, Concord, MA, USA) is a CHO-produced fusion protein containing the 110 kDa human GAA precursor and the Fab fragment of a murine lupus anti-DNA antibody, 3E10. This monoclonal anti-DNA antibody were shown to penetrate living cells and move to the nucleus through the equilibrative nucleoside transporter 2 (ENT2) [113,114,115]. Experiments in cultured L6 myoblasts and fibroblasts derived from Pompe disease patients as well as in vivo studies in KO mice suggested a potential benefit of this fusion protein [116]. However, in our experience, VAL-1221 in KO mice did not show any improvement in muscle glycogen content (unpublished data). A Phase 1/2 dose-escalation clinical trial (NCT02898753) of VAL-1221 in previously treated LOPD patients was initiated in 2017 [117], but the results of this study were not validated by peer review. As of June 2020, the company terminated this trial in the US and UK.

Finally, a combination of alglucosidase alfa with β2 agonists, clenbuterol, or albuterol, was shown to enhance the efficacy of the therapeutic enzyme owing to the increased expression of CI-MPR in skeletal muscle [118,119]. A pilot study of albuterol plus ERT in LOPD patients who were not improving further following more than two years on ERT alone showed the benefit of this approach [120]. A phase 1/2 double-blind, randomized, placebo-controlled 52-week study (NCT01942590) of clenbuterol in LOPD patients treated with ERT provided evidence for safety and showed a modest improvement in motor function [121]. This study (initiated in 2013) is now completed. Table 1 summarizes the above-mentioned clinical trials.

### 4.2. Gene Therapy

There is a dramatic surge in the number of drug candidates for gene therapy to combat human diseases, and Pompe disease is no exception. The limitations of the currently available ERT, along with the requirement for frequent life-long i.v. infusions, and the inability of the therapeutic enzyme to cross BBB, make the development of gene therapy for Pompe disease an attractive option. Over the past years, the field has witnessed the explosion of gene therapy studies testing different types of vectors, various promoters, numerous elements of the transgene expression cassette, and routes of delivery in preclinical setting. These studies are discussed in several recent reviews [102,122,123,124,125]. Here, we focused on already initiated and planned gene-therapy-based clinical trials using nonpathogenic adeno-associated virus (AAV) as a vector (Table 1). Unlike the wild type virus, the recombinant AAV genome largely remains in an episomal form in the nucleus and has a low frequency of integration in the host cell genome (reviewed in [126]). AAV quickly became the vectors of choice for Pompe disease gene therapy.

Based on the results of the early preclinical studies evaluating the effect of systemic or intramuscular administration of AAV vectors in KO mice [127,128,129], the first-in-human trial of gene therapy for Pompe disease began in 2006, thus marking a milestone in the field. This was an open label, Phase 1/2 trial (NCT00976352) using direct injection of rAAV2/1-CMV-hGAA into the diaphragm of a small group of children who required assisted ventilation despite ERT. The study confirmed safety and showed a tendency to improve respiratory function in some patients [130,131,132].

Phase 1/2 clinical trial evaluating the feasibility of two successive intramuscular (into tibialis anterior muscle) administration of an AAV9 vector expressing GAA is ongoing and is recruiting patients with LOPD (NCT02240407). The recombinant AAV carries the codon-optimized acid alpha-glucosidase under the control of human desmin enhancer/promoter (rAAV9-DES-hGAA). The immune modulation strategy using Rituximab and Sirolimus prior and after the first administration of the vector is designed to prevent the immune response against the AAV capsid and the transgene, thus allowing for the second vector administration. The same group of researchers from University of Florida are planning to initiate a new Phase 1/2 clinical trial of systemic injection of rAAV9-DES-GAA in 3–5-year-old IOPD patients.

Another approach-hepatic gene transfer-relies on a remarkable ability of hepatocytes to produce and secrete the expressed protein into the bloodstream, thus providing a steady supply of the therapeutic enzyme for the uptake by other tissues. Early studies on KO mice demonstrated that a single i.v. administration of a modified adenovirus (AV) vector encoding human GAA resulted in efficient liver transduction, secretion of the GAA precursor, and clearance of lysosomal glycogen accumulation in skeletal and cardiac muscles [133]. These results along with the follow-up studies [134,135] provided a solid foundation for using what is now called “liver depot gene therapy” in Pompe disease. However, the use of AAV vectors is now highly preferred for achieving a long-term persistent therapeutic gene transfer.

The unique immunologic properties of the liver allow for the induction of immune tolerance to foreign antigens through a regulatory T-cell mediated mechanism (reviewed in [136]). Indeed, AAV-mediated liver-specific expression of human GAA in KO mice was shown to prevent the formation of anti-GAA antibody when the vector was administered prior to the start of ERT, thus improving ERT efficacy [137,138,139]. The concept of induction of tolerance to the therapeutic enzyme delivered during ERT by low-dose AAV vector administration is termed “immunomodulatory gene therapy” [140]. Multiple preclinical studies have explored the impact of different AAV serotypes and promoters, vector dosages, and modifications of the GAA sequence, such as the signal peptide and codon optimization, to enhance GAA secretion into the bloodstream and to better control humoral immune responses (reviewed in [123,124,141]). These studies have culminated in the first liver gene therapy clinical trials for Pompe disease.

A Phase 1 clinical trial (NCT03533673) of liver depot gene therapy in adult patients is designed to evaluate an rAAV serotype 8 vector carrying the human GAA under the control of liver-specific promoter (AAV2/8-LSPhGAA). This is an ongoing open label, randomized study (currently recruiting). A careful consideration of the vector dosage for this trial was based on the preclinical data showing effective biochemical correction of skeletal muscle at a dose of 2 × 10^12^ vg/kg, and induction of immune tolerance to the ERT delivered rhGAA (with only partial correction of the muscle defect) at a minimum effective dose of 2 × 10^11^ vg/kg [142]. These data justified a starting dose of 1.6 × 10^12^ vg/kg for Phase 1 clinical trial [141]. This clinical trial was preceded by a Pompe gene therapy trial (NCT03285126; completed), designed to determine eligibility for the forthcoming trial in adults with LOPD.

Another Phase 1/2 liver transfer gene therapy clinical trial (NCT04093349; RESOLUTE), initiated by Spark Therapeutics is recruiting patients with LOPD receiving ERT. The study is designed to evaluate the safety, tolerability, and efficacy of investigational liver-directed AAV gene therapy of the secretable GAA (SPK-3006) in adults, treated in sequential dose-level cohorts. Preclinical studies using this AAV8-mediated liver gene transfer of an engineered secretable GAA (secGAA) resulted in high and stable levels of GAA in the circulation and rescued muscle and CNS pathology in adult and severely affected older KO mice without development of humoral immune responses to the enzyme [143,144].

An inherent limitation of hepatic gene transfer is that AAV episomal vectors will be diluted over time in the developing and growing liver, eventually leading to the loss of vector genomes and transgene expression. This creates a major problem for treatment of pediatric patients with the disease, who are likely to require a second round of vector administration and immunosuppression to prevent the formation of neutralizing antibodies. In general, the success of liver-directed gene therapy for Pompe disease relies on both secretion of large amount of precursor GAA and its efficient CI-MPR-mediated uptake and lysosomal trafficking in distant organs. As with traditional ERT, the same requirement for high M6P content and affinity for the CI-MPR applies to the liver-produced secreted GAA to achieve efficient targeting and correction of muscle defect. This raises a possibility of generating an “ideal” transgene expression cassette to allow for the maximally reduced dose of vector. Although challenging, the era of gene therapy for Pompe disease has arrived, and the therapy has the potential to one day become a lifelong cure.

## Figures and Tables

**Figure 1 biomolecules-10-01339-f001:**
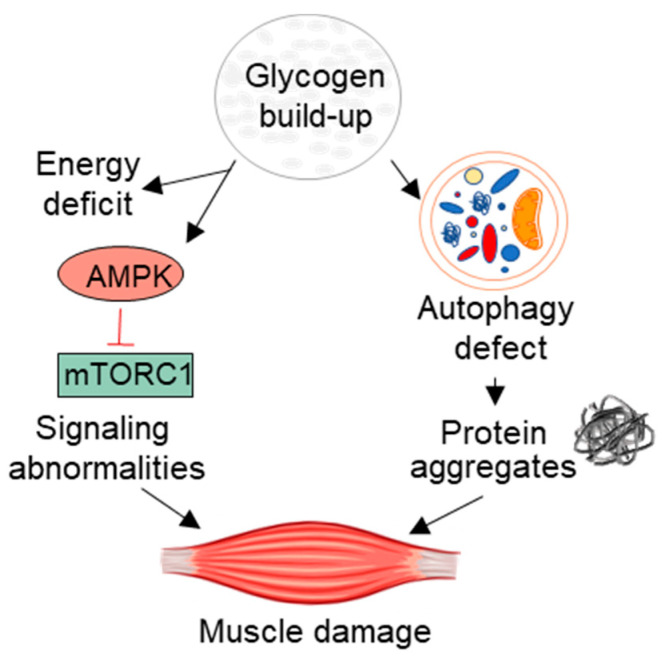
Pathogenic cascade of muscle damage in Pompe disease.

**Table 1 biomolecules-10-01339-t001:** Next generation therapies for Pompe disease.

Intervention/Treatment	Characteristics/Delivery Method	Company/Institution	Clinical Trial Phase/Identifier	References
**ERT**				
neo-GAA	Glycoengineered recombinant GAA with increased bis-M6P levels (avalglucosidase alfa)	Genzyme, a Sanofi Company	Completed/NCT01898364 Phase 3/NCT02782741	Zhu et al. [108] Pena et al. [109]
AT-GAA (ATB200/AT2221)	rhGAA bearing high bis-M6P with a pharmacological chaperone (miglustat)	Amicus Therapeutics	Phase 3/NCT03729362 Phase 3/NCT03911505	Khanna et al. [111] Xu et al. [92] Meena et al. [66]
VAL-1221	Fusion protein containing antibody 3E10 and rhGAA	Valerion Therapeutics	Terminated/NCT02898753	Weisbart et al. [114,115] Yi et al. [116] Kishnani et al. [117]
ERT + Clenbuterol	Alglucosidase alfa with β2-adrenergic agonist clenbuterol	Duke University	Completed/NCT01942590	Koeberl et al. [118,119,120,121]
**Gene Therapy**				
rAAV2/1-CMV-hGAA	Intramuscular injection into the diaphragm	University of Florida	Completed/NCT00976352	Smith et al. [130] Byrne et al. [131] Corti et al. [132]
rAAV9-DES-hGAA	Intramuscular re-administration	Lacerta Therapeutics/University of Florida	Phase 1/2/NCT02240407	Salabarria et al. [125]
AAV2/8-LSPhGAA	Screening for eligibility Ascending dose intravenous administration	Duke University Asklepios Biopharmaceutical/Duke University	Completed/NCT03285126 Phase 1/2/ NCT03533673	Kishnani et al. [141] Han et al. [142]
SPK-3006 (AAV liver directed secretable GAA)	Intravenous administration	Spark Therapeutics	Phase 1/2/ NCT04093349	Puzzo et al. [143] Cagin et al. [144]

ERT: enzyme replacement therapy; M6P: mannose-6-phosphate; clinical trials listed in this table are registered as of August 2020.

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
