# Peer review of "Pompe Disease: New Developments in an Old Lysosomal Storage Disorder"

_biomolecules, 2020, doi:10.3390/biom10091339_

Round 1

Reviewer 1 Report

The paper is a review of the current status of our understanding of the physiopathology and therapeutic options for Pompe disease, including concerns on longterm follow-up of enzyme therapy recipients, apart from implications of CRIM status. Personally, I would not favor the tile of Oldest LSD, as one can argue that other LSDs were clinically characterized earlier, although the lysosomal enzyme deficiency may have been first described (acid alpha glucosidase). The paper is well written and uptodate. However, I would request inclusion of recent studies related to biomarkers, which would be of interest to readers, with a comment of the authors thoughts regarding its potential role in patient management:

  1. Tarallo A, Carissimo A, Gatto F, et al. microRNAs as biomarkers in Pompe disease. Genet Med. 2019 Mar;21(3):591-600.
  2. Carrasco-Rozas A, Fernández-Simón E, Lleixà MC, et al. Identification of serum microRNAs as potential biomarkers in Pompe disease. Ann Clin Transl Neurol. 2019 Jul;6(7):1214-1224.

Incidentally, there is a recent review as well; but this does not give as much detail in relation to therapeutic aspects:

Taverna S, Cammarata G, Colomba P, et al. Pompe disease: pathogenesis, molecular genetics and diagnosis. Aging (Albany NY). 2020 Aug 3;12(15):15856-15874.

Author Response

We thank the reviewer for the comments and suggestions.

We agree with the reviewer that the "Oldest LSD" in the title is misleading.The title has been changed. The new title reads: "Pompe Disease: New Developments in an Old Lysosomal Storage disorder" Thank you for the correction.

We have included a paragraph on biomarkers and all three suggested references in the revised manuscript.  

Reviewer 2 Report

The authors of “New Developments in the Oldest Lysosomal Storage Disorder – Pompe Disease” provide an excellent review of the history and new updates regarding the pathological cascade in both IOPD and LOPD patients. They also give provide details of each clinical trial that explores new therapies.  This is a needed comprehensive review, which will be beneficial to the Pompe disease community.

I have just a few minor concerns:

Page 3 lines 93-112 – mention should be made of respiratory motor neuron pathology found in an ERT treated child and correlated with significant pathology in the motor neurons of the Gaa-/- mouse model (Deruisseau et al. Neural deficits contribute to respiratory insufficiency in Pompe disease; PNAS 2009)

Line 127-128 mention is made of lower airway smooth muscle pathology but this is not referenced –(Keeler et al did a comprehensive assessment of Pompe lower airway smooth muscle structure and function and should be referenced.

Prior to publication, I recommend that the manuscript be edited to comply with the gene and protein nomenclature for mouse and humans as defined by the HUGO Gene Nomenclature Committee (HGNC) & the Mouse Gene Nomenclature Committee.

Author Response

We thank the reviewer for her/his evaluation of the manuscript and we introduced the suggested changes in the revised manuscript.

Minor comments:

Respiratory motor neuron pathology and the Deruisseau et al reference have been added;

Keeler et al. reference has been added; 

The gene and protein nomenclature has been verified and updated. 

Reviewer 3 Report

Thank you for giving me the opportunity to review this interesting article by Raben et.al. The authors provided insightful explanations of the pathogenesis of muscle damage, clinical characteristics, and evolution of therapy for Pompe disease.  

  1. First to third paragraphs: Some of the sentences are lengthy and hard to read at first glance. I would recommend shortening or split these sentences. 
  2. The author and the team made great contributions to the pathogenic cascade of muscle damage. Figure 1 is helpful for readers to understand the complicated pathway. But I would recommend shortening the part of “3. Beyond the Lysosome: Pathogenic Cascade and Muscle Regeneration” into one to two pages. Or at least the author should point out the critical part in the abstract. 
  3. Line 160-163. The newborn screening for Pompe disease has been well established. However, how to diagnose and treatment of Pompe disease especially asymptomatic or mildly symptomatic cases is still challenging. When and how to treat LOPD is still unknown. Do we have any guidelines or solutions on it? 
  4. Line 343-345 Do we understand why the reduction of muscle glycogen does NOT improve in motor function? 
  5. What are the pros and cons of different kinds of treatment? Maybe worth pointing out in Table 1. 

Author Response

We thank the reviewer for careful evaluation of the manuscript. 

1/2. We prefer to keep the text as is (point 1 and point 2 of the reviewer's comments). We do indicate in the abstract that the pathogenesis of the disease will be discussed in the review.     

3. The whole paragraph at the end of Section 2 ("An expanded set ....") is on the current guidelines and on  "when" to treat LOPD.

4. The marginal improvement of motor function in older mice (Zhu et al. 2009) may result from the expanded areas of autophagic accumulation and/or neuronal involvement. However, we don't think that our speculations are appropriate. We describe the result as indicated in the referenced paper. The sentence  "...in older symptomatic KO mice..." can be removed, if that is the reviewer's preference.

5. "Pros and cons of different kinds of treatment" can easily be a subject of separate review and would be highly speculative, since most of the new therapeutic approaches are at the stage of clinical trials, and only future fill give us a definitive answer.